# A Humidity-controlled Fast Integrated Mobility Spectrometer (HFIMS) for rapid measurements of particle hygroscopic growth

Tamara Pinterich[1], Steven R. Spielman[2], Yang Wang[1], Susanne V. Hering[2], Jian Wang[1]

[1]Brookhaven National Laboratory, Upton, NY 11973-5000, USA
[2]Aerosol Dynamics Inc., Berkeley, CA 94710, USA

*Correspondence to*: Jian Wang (jian@bnl.gov)

**Abstract.** We present a Humidity-controlled Fast Integrated Mobility Spectrometer (HFIMS) for rapid particle hygroscopicity measurements. The HFIMS consists of a differential mobility analyzer (DMA), a relative humidity (RH) control unit and a water-based FIMS (WFIMS) coupled in series. The WFIMS (Pinterich et al., 2017) combines the Fast
Integrated Mobility Spectrometer (Kulkarni and Wang, 2006a, b) with laminar flow water condensation methodologies (Hering and Stolzenburg, 2005; Spielman et al., 2017). Inside the WFIMS, particles of different electrical mobilities are spatially separated in an electric field, condensationally enlarged and imaged to provide 1-Hz measurements of size distribution spanning a factor of ~3 in particle diameter, sufficient to cover the entire range of growth factor for atmospheric aerosol particles at 90 % RH. By replacing the second DMA of a traditional hygroscopicity tandem DMA (HTDMA) system
with the WFIMS, the HFIMS greatly increases the speed of particle growth factor measurement.

The performance of the HFIMS was evaluated using NaCl particles with well-known hygroscopic growth behavior, and further through measurements of ambient aerosols. Results show that HFIMS can reproduce, within 2 % the literature values for hygroscopic growth of NaCl particles. NaCl deliquescence was observed between 76 % and 77 % RH in agreement with the theoretical value of 76.5 % (Ming and Russell, 2001), and efflorescence relative humidity (43 %) was found to lie within
the RH range of 41 % to 56 % reported in the literature. Ambient data indicate that HFIMS can measure the hygroscopic growth of five standard dry particle sizes ranging from 35 to 165 nm within less than three minutes, which makes it about an order of magnitude faster than traditional HTDMA systems.

## 1 Introduction

The hygroscopicity of atmospheric aerosols is a key parameter in determining their impact on global climate. The uptake of
water by individual particles increases the light scattering, enhances heterogeneous chemical transformations important to secondary aerosol formation (e.g., Surratt et al., 2010), and is important in the formation of cloud droplets. The abundance of hygroscopic particles that act as cloud condensation nuclei affects cloud formation and cloud droplet number concentrations, which in turn influences cloud albedo, coverage and lifetime (Twomey 1977; Albrecht 1989). These "indirect effects" of atmospheric aerosols on the Earth's radiation balance remain one of the largest uncertainties in understanding climate change

(IPCC 2013). Hygroscopicity is among the key determinants of the ability of aerosol particles to form cloud droplets and therefore the aerosol indirect effects (e.g. Mei et al., 2013; Liu and Wang, 2010).

Most commonly particle hygroscopic growth is measured using hygroscopicity tandem differential mobility analyzer (HTDMA) systems, which consist of two differential mobility analyzers (DMAs) in series, separated by a means to control

the sample flow relative humidity (RH). HTDMA systems first select a single particle size using the first DMA, change its relative humidity environment, then scan the classifying voltage of the second DMA to measure the distribution of particle sizes resulted from the change in RH. The HTDMA method is accurate, but slow. Typically the time required to complete a measurement cycle of determining the growth factor at a single relative humidity (such as 90 %) for 5 different particle sizes is about 30 min (e. g. Cerully et al., 2011). Measurement periods are especially long for large particles which are low in

concentration, and small particles that have a low charging efficiency.

Several investigators have worked to increase the speed of HTDMA measurements by replacing the second DMA with an instrument that is capable of fast size distribution measurements. Sorooshian et al. (2008) developed a Differential Aerosol Sizing and Hygroscopicity Spectrometer Probe (DASH-SP), in which wet particle size is measured by an optical particle counter (OPC). By replacing the 2nd DMA with an optical counter, DASH-SP accelerates the measurement significantly.

However, the optical counting limits DASH-SP measurements to particles larger than ~150 nm in diameter, and a sophisticated algorithm is required to account for the variation of particle refractive index due to water uptake and its impact on optical sizing. Stolzenburg et. al (1998) developed a high-flow mobility analyzer which they coupled to an optical particle counter, with humidity control upstream of the mobility size separation, and an aerosol dryer downstream to measure particle size change upon dehumidification. Leinert and Wiedensohler (2008) developed a DMA-aerodynamic particle sizing (APS)

system to examine growth factors, but their measurements were complicated by the change in particle density, which affects the aerodynamic measurement. While faster, the limitation of these coupled DMA-optical counter or DMA-aerodynamic sizing techniques is the limitation on the measurement size range and the additional complexity or uncertainty in mapping the optical or aerodynamic size onto the physical size of the particle that is important to assessing water uptake. These systems, based on optical sizing or aerodynamic sizing, are capable of measuring the hygroscopicity of large accumulation

mode particles, which are important to evaluating the optical properties and direct radiative effects of ambient aerosols.

To address the need for fast and precise measurements of particle hygroscopic growth, we have developed a Humidity-controlled water-based Fast Integrated Mobility Spectrometer (HFIMS), which replaces the second DMA of the HTDMA systems with a water-based FIMS (WFIMS; Pinterich et al., 2017). By detecting particles of different sizes simultaneously, WFIMS provides rapid measurements of the size distribution of humidified particles. Unlike the final optical sizing of

Sorooshian et al. (2008), or final aerodynamic sizing of Leinert and Wiedensohler (2008), WFIMS measures particle sizes based on electrical mobility. This removes the uncertainty introduced by particle refractive index or density, and provides the same, precise growth factor measurements of the HTDMA systems, but with a much faster measurement speed. Compared to systems based on optical or aerodynamic sizing, HFIMS extends fast measurements to particles with diameters below 150 nm. Particles smaller than 150 nm often represent a large fraction of cloud condensation nuclei population, and may have a

strong impact on human health (Chen et al. 2016). Using both laboratory and ambient measurements, we demonstrated that the HFIMS can provide growth factor measurements with 1 % precision for five representative particle diameters in less than 3 minutes - about one order of magnitude faster than traditional HTDMA systems.

## 2 Instrument design

The HFIMS consists of three individual units (see Fig. 1): a TSI DMA (either long-column or nano-column, depending on the particle size) classifying particles at a desired dry size under a low RH, an RH control unit providing independent controls of the size-selected particle sample and WFIMS sheath flow RH, and a WFIMS measuring size distributions of particles after being exposed to a different RH. The WFIMS used here is identical to the original WFIMS (Pinterich et al., 2017), except the HV electrode is replaced with one that provides a uniform electrical field with a small offset from aerosol

inlet slit, as described below. This modification is made to optimize the measurements of humidified particle size distributions. In essence, the WFIMS deployed in this study is similar to the alcohol based FIMS reported in Kulkarni and Wang (2006a), except that particle growth is achieved by condensation of water instead of butanol, which is key to hygroscopicity measurements.

### 2.1 Relative humidity control (RH-Control)

An automated RH control system was constructed to independently control the RH of the size selected aerosol sample flow ($RH_a$), and the WFIMS sheath flow ($RH_{sh}$). Humid air (>95 % RH) is created by bubbling dry air through sintered metal mufflers submerged in about 20 cm of water. The humid air is then mixed with dry air to provide WFIMS sheath flow. The RH of sheath flow is controlled by a PID controller that drives a proportional solenoid valve on the dry air line based on the sheath flow RH probe reading. The aerosol sample flow RH is controlled using a Nafion® exchanger, and the dry-humid

mixture used as the purge flow of this exchanger is controlled independently. As with the sheath flow, the purge flow is obtained by mixing of dry air with that from the humid air source, with a second PID controller that reads the aerosol RH and drives the valve on the dry air line to meet the target aerosol sample RH. To compensate for the evaporative cooling, the bubbler is equipped with a heater and a simple thermostat, set to the room temperature (not shown in Fig. 1).

### 2.2 Configuration of the WFIMS

The WFIMS is identical to the original version presented in Pinterich et al. (2017) except that the HV electrode is replaced with one that provides a uniform electric field with a slight offset from aerosol inlet slit. The WFIMS consists of a parallel plate mobility separator followed by a three-stage condensational growth channel and an imaging system. A particle free sheath flow $Q_{sh}$ enters at the top, and an aerosol flow $Q_a$ is introduced through a slit along the entire width of the separator channel. A constant total flow $Q_{tot}$ of 16.5 l/min through the channel is achieved by a vacuum pump along with a critical

orifice (O'Keefe Controls Co., No. 55). The desired $Q_a$ of 0.3 l/min is achieved via PID control of $Q_{sh}$ using a flow control

valve (MKS, 0248A). The key physical dimensions and operating conditions of the WFIMS in this study are listed in the supplement Sect. S1.

The WFIMS is configured with a single voltage electrode that has an offset in the direction of the flow ($z$-direction). A side-view (($x,z$) - plane) of the WFIMS separator is shown in Fig. 2, where the flow is downward, and the high voltage electrode on the right extending in $y$-direction. For measurements of particle growth factor (GF), i.e. the ratio of humidified to dry particle diameter $D_p/D_{p,0}$, the size range of the WFIMS only needs to cover the possible change in particle diameter. For ambient particles, the growth factor reported in the literature ranges from 0.8 to 2.2 for RH values up to 90 %. This range covers the GF of sea salt like particles (Ming and Russell, 2001), and also encompasses the 15 % shrinkage observed for highly agglomerated particles that occurs when the branched structure collapses following the capillary condensation of water (Weingartner et al., 1995). Conveniently, WFIMS operated with a single-voltage electrode has an electrical mobility range of a factor of 10 (Kulkarni and Wang, 2006a, b), which corresponds to more than a factor of 3 in particle diameter, sufficient to cover the GF range of atmospheric aerosols.

The single high voltage electrode was configured with an offset such that the high voltage region (red area in Fig. 2) begins slightly downstream (32 mm) of the introduction of aerosol into the separator. The offset allows the sheath flow to provide the additional RH conditioning of the size selected sample flow prior to mobility classification as the aerosol flow is less than 2 % of the total flow. As the control of $RH_a$ is achieved by using a Nafion exchanger and has a longer response time than that of WFIMS sheath flow, this feature could accelerate the growth factor measurements under different RH by reducing the waiting time following the change of RH setpoints. However, in the present study we mainly focus on measurements at a single RH (85 %) where this feature is not relevant.

A heater and thermistor were attached near the bottom of the separator to compensate the heat loss to the adjacent cooled conditioner stage (see details in the next paragraph) and to avoid a corresponding change in RH due to this gradient. The heater was driven to equalize the temperatures within the separator. Without heating, the temperature at the bottom of the separator is about 1.0 °C below that at the top.

Upon exiting the separator the particles continue along their flow trajectories through the three stage growth channel, consisting of conditioner, initiator and moderator, all with wetted walls (Spielman et al., 2017). Particles are enlarged through water condensation without being diverted from their trajectories. WFIMS' three-stage growth channel design provides supersaturation levels of ~1.35 across its viewing window, sufficient to activate and grow particles as small as 7 nm to detectable sizes (Pinterich et al., 2017), while also removing excess water vapor that might otherwise condense on the optical components (Hering et al., 2014). Within the final section, grown particles are illuminated by a laser beam and imaged by a camera at a frame rate of 10 Hz. MATLAB's "Image processing toolbox" is used to detect each droplet and its position. Only particles detected in the center of the channel cross section (($x,y$) - plane) are used for measurements of particle size and concentration in order to avoid the edge effects of electric and flow fields (Olfert et al., 2008). For WFIMS operated with a single voltage electrode, particle positions can be converted to instrument response mobilities $Z_p^*$, using Eq. (22) from Kulkarni and Wang (2006a)

$$Z_p^* = \frac{2(1+\beta)(3\tilde{x}^2 - 2\tilde{x}^3) - \beta}{2+\beta} \cdot Z_{p,max}^* \tag{1}$$

with flow ratio $\beta = Q_a/Q_{sh}$ and $\tilde{x}$ being the ratio of the $x$-coordinate of the detected particle to separator gap width $a$. Note the conversion of particle position into instrument response mobility is independent of the applied particle enlargement technique (alcohol vs. water). The maximum response mobility $Z_{p,max}^*(\tilde{x} = 1)$ measured by the (W)FIMS can be expressed as (Kulkarni and Wang 2006a):

$$Z_{p,max}^* = \frac{a \cdot Q_a}{\beta(b \cdot l_s \cdot V)} \tag{2}$$

with $b$ and $l_s$ being electrode width and length, respectively. $V$ is the voltage applied across the high voltage electrode. Knowing $Z_{p,max}^*$ the mobility resolution $R$, which is defined as the ratio of particle mobility $Z_p$ to the full width at half height of the WFIMS transfer function $\Delta Z_p^*$, can be calculated according to (Kulkarni and Wang, 2006a)

$$R = \frac{Z_p}{\Delta Z_p^*} \cong \frac{Z_p}{\beta \cdot Z_{p,max}^*}. \tag{3}$$

Similar to a DMA, the mobility resolution of the (W)FIMS depends on the ratio of sheath to aerosol flow. In addition, (W)FIMS resolution is also a function of particle mobility $Z_p$. In DMA classifiers all selected particles traverse the entire mobility separation channel. In contrast, in (W)FIMS systems, only the most mobile particles traverse the entire channel, while the less mobile particles traverse just a portion of the sheath flow. Thus in the (W)FIMS, the mobility resolution reaches the maximum value of $1/\beta$ only for largest mobility measured (i.e., $Z_{p,max}^*$) while the resolution for less mobile particles is lower. When operated at a fixed sheath flow of 16.2 l/min, and an aerosol flow of 0.3 l/min, as is our standard operating configuration, the resolution for those particles that traverse the entire channel is 54, while that for particles traversing just one-half of the channel will be 27.

In this study, WFIMS separating voltages ranged from 70 to 4500 V, allowing hygroscopicity measurement for particles with dry diameters ranging from 15 to 205 nm. This includes the standard sizes from 35 to 165 nm suggested by the EUSAAR project (Duplissy et al. 2009). Dry particle size ranges of the HFIMS and other representative instruments are shown in Table 1. Compared to HFIMS, systems based on optical or aerodynamic sizing have a larger upper size limit. On the other hand, HFIMS is capable of rapid measurement of particles with dry diameters below 150 nm, which are difficult to detect using OPC or APS.

**3. Experimental setup**

The capability of HFIMS to accurately characterize particle hygroscopicity is examined by measuring sodium chloride particles, for which hygroscopic growth has been well characterized in prior studies. The experimental setup is shown in Fig. 3. NaCl particles were generated by atomizing a dilute NaCl solution (1.7 mM) using a constant output atomizer (TSI Inc., Model 3076), followed by a diffusion dryer. A stable particle number concentration of about 50 cm$^{-3}$ was achieved by adapting (i) the diameter of a limiting orifice in dilution stage 1 and (ii) the dry, particle free dilution flow via a needle valve

(dilution stage 2). After dilution polydisperse particles were charge equilibrated using a $^{85}$Kr bipolar aerosol neutralizer (TSI Inc., Model 3077) and size selected at low humidity (RH < 10 %) using a nano-column DMA (TSI Inc., Model 3085). In the present study we selected 50 nm particles. The ratio of DMA aerosol to sheath and sample to excess flow was kept constant at 1:10. During laboratory characterization a CPC (TSI Inc., Model 3010) was operated downstream of HFIMS' DMA in parallel to the RH-Control – WFIMS branch, and it provided the concentration of size selected dry particles. WFIMS was operated at constant separating voltage $V$ of 1000 V corresponding to a growth factor window of 0.9 to 3.2. A detailed list of HFIMS configuration parameters during the laboratory characterization can be found in the supplement Sect. S2. Note due to non-idealities of the electric field, instrument response mobilities $Z_p^*$ were calculated using Eq. (1) with an effective voltage $V_{eff}$ instead of the applied voltage $V$ (Eq. (2)). The $V_{eff}$ was derived following a calibration procedure detailed in the supplement Sect. S3.

For the measurements of deliquescence and hygroscopic growth of NaCl particles, we matched the relative humidity of aerosol and sheath flows ($RH_a = RH_{sh}$). The measurement of efflorescence was carried out by keeping $RH_a$ at 85 % while varying $RH_{sh}$ between 18.8 % and 79.9 %.

The measurement speed of HFIMS was evaluated by sampling ambient aerosols outside of our laboratory at Brookhaven National Laboratory (Upton, New York). Figure 4 shows the schematic of the experimental setup. We obtained ambient particle growth factors (GF) at $RH_a = RH_{sh} = 85$ %, for various dry particle diameters including five diameters (i.e., 35, 50, 70, 110 and 165 nm) used by EUSAAR (European Supersites for Atmospheric Aerosol Research; Duplissy et al., 2009). A detailed list of HFIMS configuration parameters during ambient tests can be found in the supplement Sect. S4. Ambient particles with dry diameter up to 110 nm were classified with a nano-column DMA (TSI Inc., Model 3085), above 110 nm a long-column DMA (TSI Inc., Model 3081) was used. Whereas different DMAs were used to classify particles with different diameters in these preliminary tests, future systems will consist of a single DMA. As a result, the time required to switch between nano- and long-column DMA was not considered when evaluating HFIMS' measurement speed. As the classified dry particle diameter varied, WFIMS separating voltage $V$ and hence mobility diameter window were adapted accordingly, and the values are listed in the supplement Sect. S4.

## 4. Results and discussion

### 4.1 Laboratory evaluation with NaCl particles

Growth factors of 50 nm NaCl particles measured under both increasing (i.e., deliquescence branch, red diamonds) and decreasing (efflorescence branch, blue circles) RH conditions are shown in Fig. 5. Measurements of deliquescence branch were carried out with matching aerosol and sheath RH (i.e., $RH = RH_a = RH_{sh}$). To account for the cubic shape of dry NaCl particles we used a shape correction factor of 1.08 (Zelenyuk et al., 2006). Hence measured mobility equivalent diameters were decreased by about 4 % to obtain volume equivalent diameters.

The NaCl deliquescence transition observed by HFIMS is just over 76 %, in agreement with the theoretical value of 76.5 % (Ming and Russell, 2001), and measurements by Hämeri et al. (2001), and Cruz and Pandis (2000) of 76 % and 75.6 %, respectively. It should be mentioned that around the deliquescence transition two distinct size modes are observed (see Fig. 6). This suggests some heterogeneity in the RH of aerosol sample (i.e., some particles experienced slightly higher RH than others), which is likely due to temperature variations among different parts of the system. Deliquescence transition data shown in Fig. 5 represent the number weighted mean growth factor for the two modes. Improved RH and temperature control could minimize the RH heterogeneity, and will be a topic of future study. Above the deliquescence transition, growth factors measured by the HFIMS are within 2 % of theoretical values, suggesting the RH heterogeneity has negligible impact on measured particle growth factors above deliquescence RH (e. g. at 85 %).

Figure 5 also shows the efflorescence curve (blue circles), that is the size change when the relative humidity environment decreases. Data were obtained by maintaining $RH_a$ constant at 85 % while varying $RH_{sh}$ between 18.8% and 79.9 %. Hence the sheath flow was used to condition the sample RH, a feature made possible by the new offset electrode (see Sect. 2.2). As demonstrated in the supplement Sect. S5 the RH of the mixed flow reaches the average value very quickly, such that the RH becomes uniform at the start of the electric field when the offset electrode is used. Humidified particle diameters $D_p$ were measured at RH given by:

$$RH = (Q_a \cdot RH_a + Q_{sh} \cdot RH_{sh})/(Q_a + Q_{sh}). \hspace{2cm} (4)$$

It is common for salt aerosols to exhibit this type of hysteresis, with the droplet becoming supersaturated while gradually decreasing in size until finally recrystallizing at a much lower relative humidity. We observed recrystallization near 43 % RH, which is close to the value of 44 % reported by Biskos et al. (2006), and within the range of 41 % to 56 % (yellow area) reported in the literature.

Using the HFIMS operating conditions listed in the supplement (Sect. S2) we calculated the resolution of its sizing unit, i.e. the WFIMS, as a function of hygroscopic growth factor for non-diffusing particles and compared it to the typical resolution of a DMA, i.e. 10. As shown in Fig. 7 the mobility resolution of HFIMS is equal to, or exceeds that of the HTDMA over the measured growth factor range, i. e. 1 – 2.27, shown in Fig. 5.

**4.2 Ambient particle hygroscopicity measurement**

Characterization of ambient aerosol hygroscopicity often requires measurements at multiple particle sizes within a reasonable time. This is often challenging for measurements using traditional TDMA systems, especially for the larger particles which are low in number concentration, and the smallest particles which have low charging efficiency.

Figure 8 shows results from ambient measurements with HFIMS, where we evaluated the relative standard error of the mean growth factor (SEM of $\overline{GF}$) as a function of particle counts $C$, and corresponding sample duration. Note as the mean particle growth factor is given by the ratio of wet to dry particle diameter, $\overline{D_p}$ and $D_{p,0}$, respectively, the relative precision in measured $\overline{GF}$ can be approximated by the relative precision in measured $\overline{D_p}$ since

$$\frac{\text{SEM}(\overline{\text{GF}})}{\overline{\text{GF}}} = \sqrt{\left(\frac{\partial \overline{\text{GF}}}{\partial \overline{D_p}}\right)^2 \cdot \text{SEM}^2(\overline{D_p}) + \left(\frac{\partial \overline{\text{GF}}}{\partial D_{p,0}}\right)^2 \cdot (\Delta D_{p,0})^2} \Big/ \overline{\text{GF}} \approx \frac{\text{SEM}(\overline{D_p})}{\overline{D_p}}. \tag{5}$$

Total particle concentrations were relatively low, around 2200 cm$^{-3}$, similar to what might be expected for continental background. The standard error of the mean (SEM) for $\overline{D_p}$ is estimated as:

$$\text{SEM}(\overline{D_p}) = \frac{\text{SD}}{\sqrt{C}} \tag{6}$$

where $C$ is the number of particles counted and SD the sample standard deviation given by

$$\text{SD} = \sqrt{\frac{\sum_{i=1}^{C}(D_{p,i} - \overline{D_p})^2}{C}} \tag{7}$$

with $D_{p,i}$ being the diameter of the $i$-th particle measured. The relative SEM of $\overline{\text{GF}}$ can be derived by combining Eqs. (5) and (6):

$$\frac{\text{SEM}(\overline{\text{GF}})}{\overline{\text{GF}}} = \frac{\text{SD}}{\overline{D_p}\sqrt{C}}. \tag{8}$$

Note while the sample standard deviation SD, which is a measure of the dispersion of measured particle sizes from the sample mean diameter $\overline{D_p}$, asymptotically reaches the population standard deviation as $C$ increases (see Figure 8 (c)), SEM, which is a measure for the uncertainty of the sample mean growth factor around the population mean, approaches zero with increasing sample size (see Figure 8 (b)) a behavior also known as law of large numbers. For narrow particle number size distributions (SD < 0.2·$D_p$) the number of particles required to reach better than 1 % precision in growth factor is on the

order of 100. The required time to detect this many particles, from now on referred to as minimum sample duration, ranged from about 100 s at 15 nm, to less than 30 s between 35 and 165 nm.

Figure 9 shows average growth factors of ambient aerosol (red circles) and corresponding minimum sample duration (black crosses) measured at constant RH of 85 % on December 3$^{rd}$, 2015. Larger particles ($D_{p,0} \geq 110$ nm) were observed to be more hygroscopic ($\overline{\text{GF}} > 1.33$). This indicates that larger particles originated from regional background aerosol, which had

been processed during transport resulting in a higher soluble fraction (Swietlicki et al., 2008). Below 110 nm particles were found to be nearly hydrophobic ($\overline{\text{GF}} = 1.0 - 1.11$). In this size range particles were probably dominated by freshly emitted combustion particles containing soot and water-insoluble organic compounds (Weingartner et al., 1997) as aerosol was sampled near a parking lot.

In addition to the average growth factor, the GF distribution of the humidified aerosol, its width, and whether it is unimodal

or bimodal are examined. Figure 10 compares size distributions of the humidified aerosol obtained by HFIMS at five particle sizes recommended by the EUSAAR project for 20 s (black), and after 200 s (red) of sampling time. All data are for size-selected, initially dry ambient particles humidified to 85 % RH. Note that the short counting times reproduce the final GF distribution nicely. Widths of the distributions are visually identical. Both short and long sample durations reveal a more hygroscopic mode appearing at 35 nm, growing in prominence at 70 nm, and becoming dominant above 110 nm.

These analyses indicate that the hygroscopic growth factors at the 5 particle sizes could be captured by the HFIMS within 3 minutes, including 15 seconds waiting time to ensure the system reaches steady state following the switching between different $D_{p,0}$. Similar measurements using conventional HTDMA systems often take about 30 minutes or more, therefore HFIMS represents an order of magnitude improvement in the measurement speed.

## 5. Conclusions

We present a Humidity-controlled water-based Fast Integrated Mobility Spectrometer (HFIMS) for rapid measurement of particle hygroscopicity.

The HFIMS consists of a DMA, an RH-control unit and a Water-based Fast Integrated Mobility Spectrometer (WFIMS) (Pinterich et al., 2017). The WFIMS combines a single high-voltage FIMS (Kulkarni and Wang, 2006a) with the laminar flow water condensation methodologies developed by Hering and coworkers. By detecting particles of different sizes simultaneously, the WFIMS provides rapid mobility based measurements of particle size distributions over a factor of 3 or more in particle diameter, which is sufficient to cover the entire range of growth factor for ambient aerosol particles. Thus, with the combination of DMA, relative humidity control, and WFIMS, the HFIMS can capture the complete growth factor distribution of size selected particles with much improved speed.

Laboratory experiments with NaCl particles showed that HFIMS can reproduce theoretical growth factors within 2 %. The deliquescence transition was observed just over 76 %, in excellent agreement with the theoretical value of 76.5 % (Ming and Russell, 2001). The measured efflorescence relative humidity (43 %) was found to lie within the range of 41 % to 56 % reported in the literature.

The hygroscopicity of ambient aerosols was characterized by keeping sample and sheath RH at 85 % and varying the dry particle size. We found that growth factors of ambient particles ranging from 35 nm to 165 nm could be measured within less than 3 minutes, providing approximately a factor of ten increase in the time resolution. The system will greatly improve our capability to study particle hygroscopic growth, especially for rapidly evolving aerosol populations.

### Acknowledgments

We thank Andrew McMahon for his help with the development of the offset high voltage electrode. This work was supported by the US Department of Energy, Office of Science, Small Business Technology Grants DE-SC0006312 and DE-SC0013103.

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

**Figures**

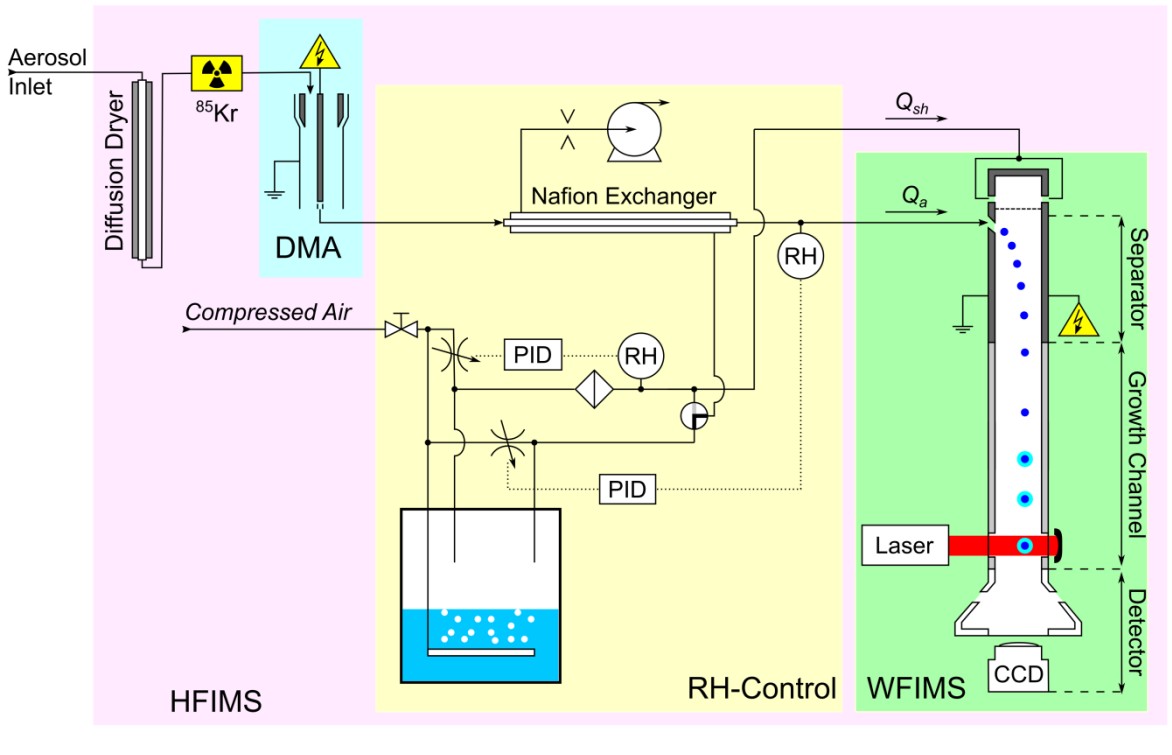

Figure 1 Schematic diagram of the HFIMS.

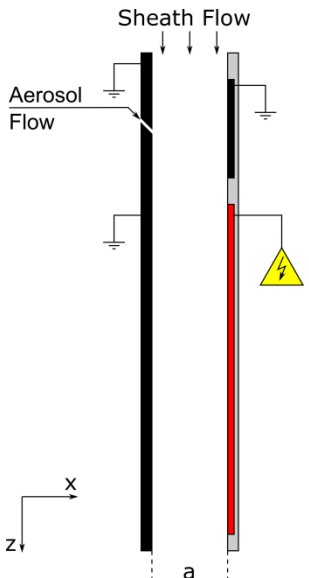

5    Figure 2 Schematic diagram of the offset electrode used in HFIMS with aerosol inlet on left.

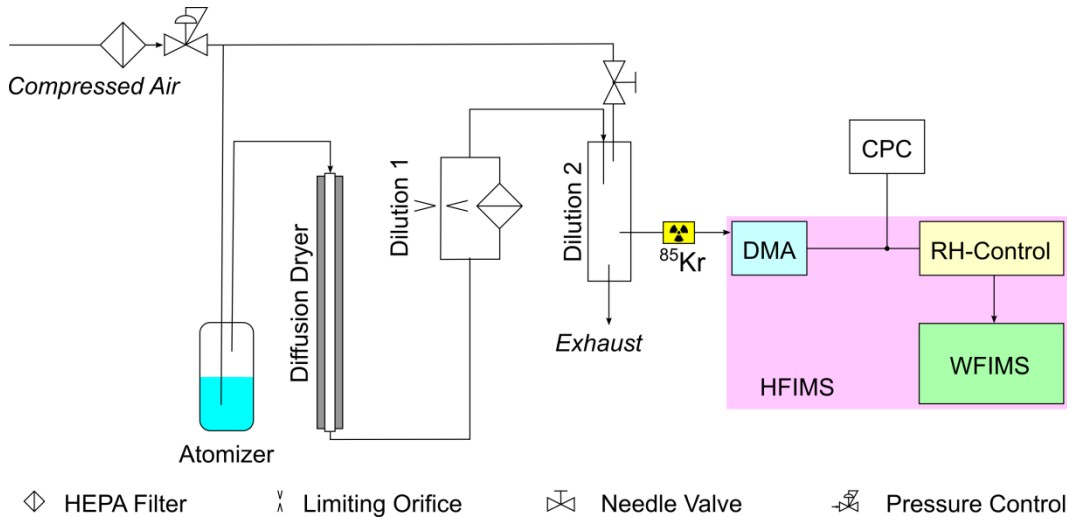

**Figure 3 Experimental setup for laboratory characterization of HFIMS.**

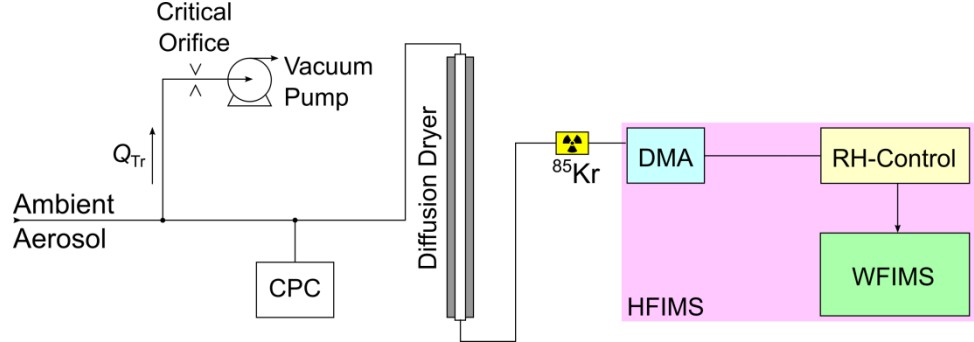

**Figure 4 Experimental setup for measuring hygroscopic growth of ambient particles.**

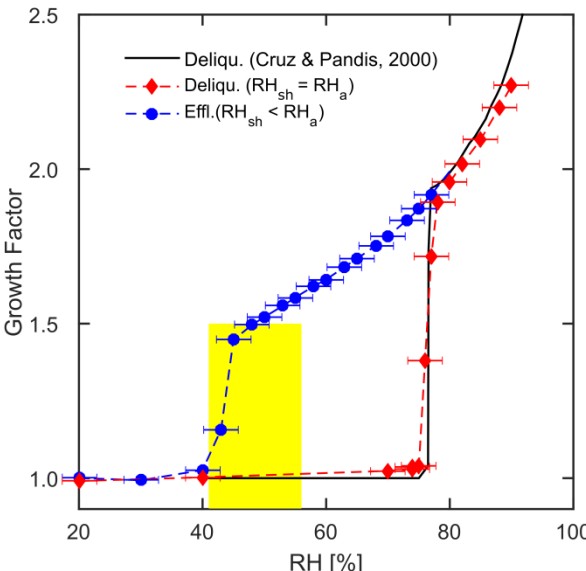

**Figure 5 Mean growth factor of NaCl particles ($D_{p,0}$ = 50 nm) measured by the HFIMS as a function of relative humidity (RH) for both deliquescence (red diamonds) and efflorescence (blue circles) branches. Horizontal error bars indicate the RH sensor (Vaisala, Model HMP60) accuracy of ±3%. Vertical error bars, which represent the standard error of the mean growth factor, are covered by the data point symbols. The solid (black) line represents NaCl ($D_{p,0}$ = 100 nm) deliquescence curve reported by Cruz and Pandis (2000). The yellow area indicates the range of efflorescence transitions reported in literature.**

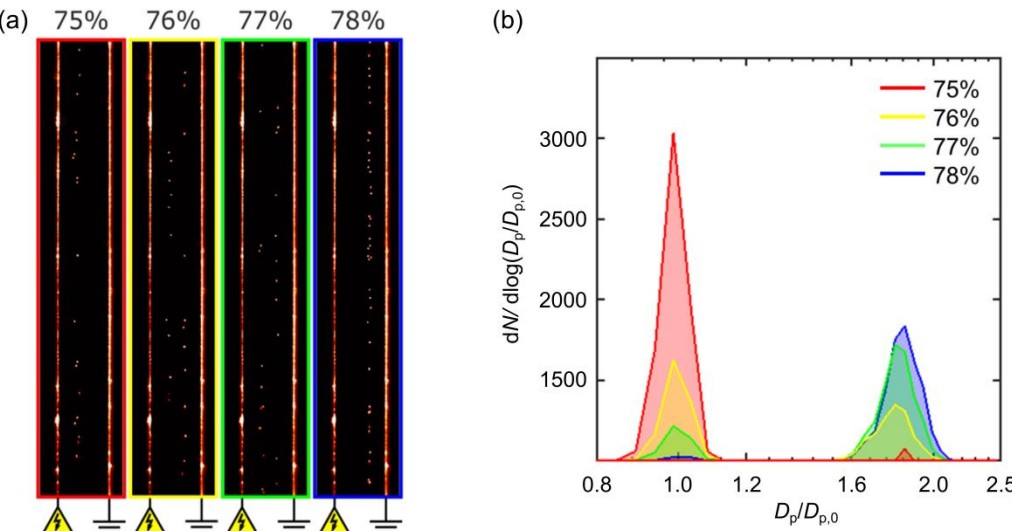

**Figure 6 (a) CCD images showing particle positions and (b) number size distributions of NaCl particles ($D_{p,0}$ = 50 nm) measured around the deliquescence point. Red, yellow, green and blue distributions correspond to 75 %, 76 %, 77 % and 78 % RH, respectively.**

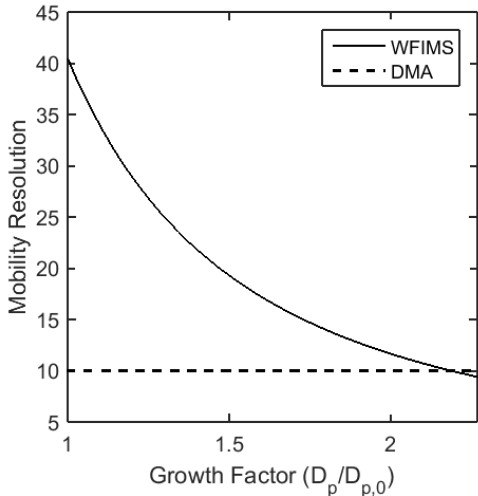

**Figure 7 WFIMS mobility resolution (solid line) and typical DMA mobility resolution (dashed line) as a function of growth factor ($D_p/D_{p,0}$) for non-diffusing particles with 50 nm dry size.**

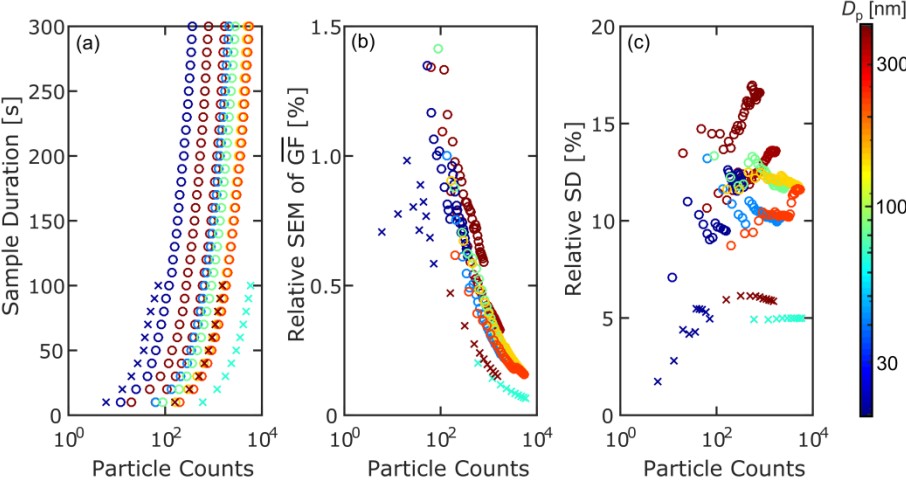

**Figure 8 (a) Sample duration, (b) relative standard error (SEM) of the mean particle growth factor $\overline{GF}$ and (c) relative standard deviation (SD) of the sample particle size distribution as a function of total particle counts for the ambient aerosols sampled on November 6 (crosses) and December 3 (circles) 2015. Colour coding represents mean particle wet sizes at RH = 85 %.**

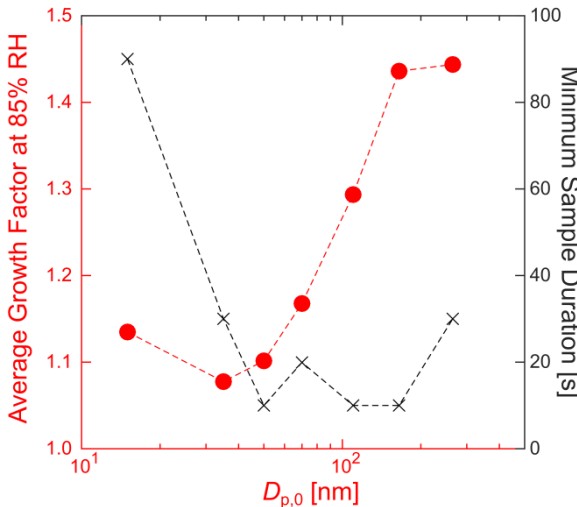

**Figure 9. Size dependent average growth factor at RH = 85 % (red circles) and corresponding minimum sample duration (black crosses) of ambient particles sampled on December 3rd, 2015.**

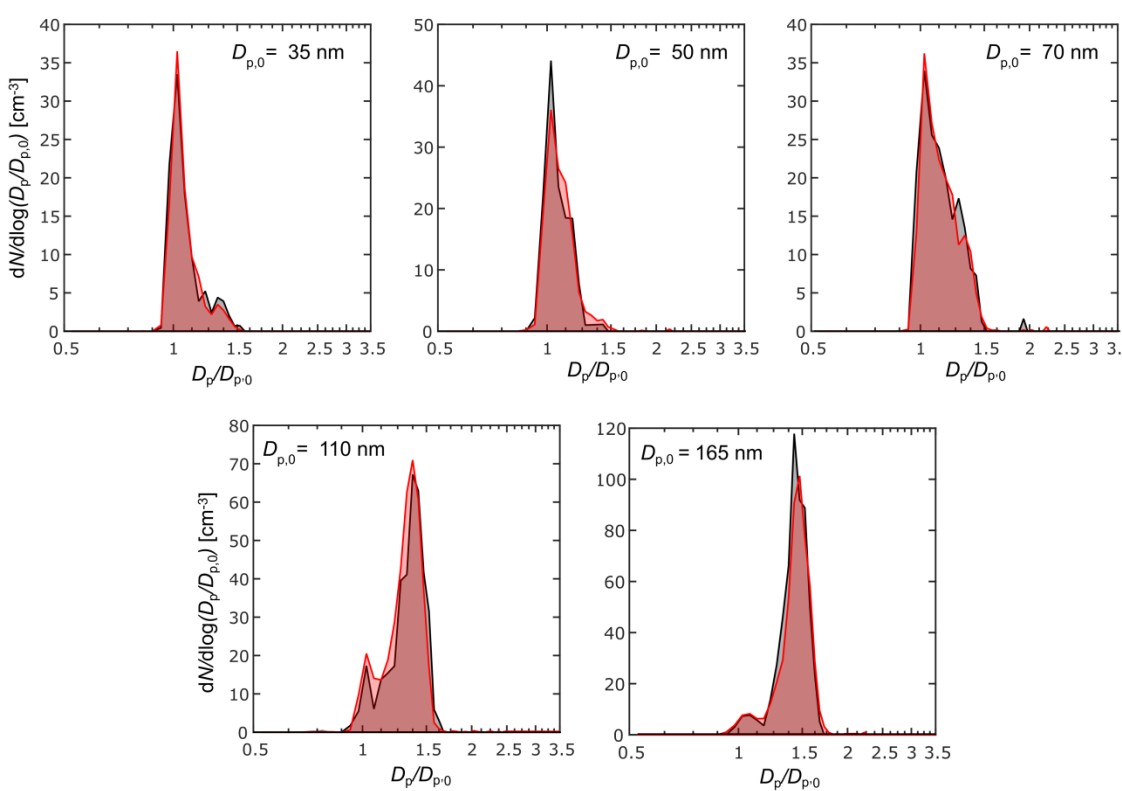

5   **Figure 10 Comparison of GF distributions of size-selected ambient particles humidified to 85 % for short (20 s, black) and long (200 s, red) sampling periods.**

**Tables**

**Table 1 Dry particle diameter range ($D_{p,0}^{min}$ - $D_{p,0}^{max}$) of four particle hygroscopicity instruments including HFIMS. Diameter ranges of HFIMS, BMI HTDMA and DASH-SP were estimated based on growth factors ranging from 0.8 to 2.2. DASH-SP dry size range was estimated for 5 l/min DMA sheath flow rate, 281 V minimum classifying voltage and 135 nm cut size of the OPC as specified in**
5 **Sorooshian et al. (2008). The particle dry diameter range of BMI HTDMA was calculated for a typical fieldwork configuration (i. e. $Q_a$ = 0.6 l/min, $Q_{sh}$ = 6 l/min, Lopez-Yglesias et al., 2014) and classifying/scanning voltages ranging from 20 V to 6000 V. The diameter range of H-DMA-APS is listed as specified in Leinert & Wiedensohler (2008).**

| Instrument | $D_{p,0}^{min}$ [nm] | $D_{p,0}^{max}$ [nm] |
|---|---|---|
| HFIMS (This study) | 15 | 205 |
| BMI HTDMA (Lopez-Yglesias et al. 2014) | 33 | 425 |
| DASH-SP (Sorooshian et al. 2008) | 170 | 455 |
| H-DMA-APS (Leinert & Wiedensohler 2008) | 800 | 1600 |

