# Peer review of "A Humidity-controlled Fast Integrated Mobility Spectrometer (HFIMS) for rapid measurements of particle hygroscopic growth"

_Atmospheric Measurement Techniques, 2017_

## Referee Comment (RC1) · Anonymous Referee #1 · 28 Jun 2017

This work reports on a new way to quantify aerosol hygroscopic growth factors. The authors introduce the Humidity-controlled water-based Fast Integrated Mobility Spectrometer (HFIMS). The topic of this paper certainly fits in this journal and the paper is very well-written and to the point. The new instrument is a good addition and highly relevant for ambient applications that require fast time resolution. The instrument provides some advantages over other recent instrument designs that conduct growth factor measurements. Characterization tests were performed that show that the instrument can accurately quantify growth factors and deliquescence/efflorescence RH values for a known inorganic salt. The authors can strengthen their manuscript by demonstrating that the instrument works well on a mobile platform. If this is not possible for some

reason, the authors should mention why this is not possible; from what i can tell, the field data shown were from a stationary set-up.

Specific Comments:

Figure 2: "Sheat flow" spelled wrong (Sheath)

The authors should provide a clear statement about the range of dry diameters the instrument can handle for the growth factor measurement. is it 35-165 nm as Lines 6-7 suggest on page 6? state it clearly and compare it to other available instruments. it is nice that it can detect smaller sizes where there may be higher concentrations, but it may be useful to point out that larger sizes may be more relevant for light-scattering and radiative forcing? For what purposes are smaller sizes most relevant (e.g., health effects)?

A few times it is stated that "...size distribution spanning a factor of $\sim$3 in particle diameter". Rather than using the confusing "factor of $\sim$3" everywhere, can the authors simply just state the range of diameters?

What is the residence time of aerosol in different parts of the instrument?

How long does it take to scan through a range of different relative humidities? Provide numbers and compare to other instruments.

What is the range of relative humidities the instrument can handle?

Section 2.1: Can the authors comment on hysteresis effects that are common with nafion based systems?

Has the instrument been deployed on a mobile platform? The paper states that the instrument is ideal for mobile platform work but it is unfortunate that data does not appear to be shown from its use in the field to give readers a better sense of how robust it is when the conditions aren't 'easy' such as in a lab or at a stationary field site.

Figure S2: x-axis has "distance" misspelled.

[Figure]

---

## Referee Comment (RC2) · Anonymous Referee #2 · 28 Aug 2017

A new instrument for fast measurement of particle hygroscopicity, HFIMS, is reported in this paper. Different from the existing fast hygroscopicity measurement instruments, HFIMS measures the electrical mobility diameters of the grown particles so that particle density and refractive index are not needed anymore. Compared with the HTDMA, HFIMS avoids the diameter scan which is time consuming therefore has a much higher time resolution and may work on mobile platforms. The topic fits well in AMT and the manuscript is well written. I therefore recommend the final publication of this paper on AMT.

Specific comments:

[Figure]

What is the maximum RH the system can reach? How about its stability at different RH levels? Since the RH is only measured before the flow entering WFIMS (not like HTDMA in which RH is also measured at the sheath exit), the real RH the particles undergo may be slightly different from the value given by RH sensors (e.g. in case of high particle concentration and very slow growth, or temperature difference between parts in the system). Does the author have any comment on this?

What is the time needed to washout all particles in the system at a typical setting of flow rates? The washout time may strongly influence the time resolution of the measurement.

The finite-width transfer function of WFIMS may cause a smooth effect in the measured distribution of GF. The discussion about it (P7L10) is a bit too brief. Can the author give a more detail discussion on the uncertain of GF distribution due to this effect? Is it possible and necessary to correct this smoothing effect?

Fig. 5: It seems that the blue dots are on the extension of the deliquescence part of the solid line (reported by Cruz and Pandis, 2000) but the red dots are a bit lower. Does the author have any explanation on this difference? Is it possible to add error bars as measurement uncertainty?

P5L6: It will be clearer if the definition of mobility resolution R is also given here.

Fig. 7: What is growth factor resolution, the ratio between growth factor and its uncertainty? It is better to give a clear definition in the text.
* * *

---

## Author Comment (AC1) · 16 Oct 2017

**Comment on "A Humidity-controlled Fast Integrated Mobility Spectrometer (HFIMS) for rapid measurements of particle hygroscopic growth" by Tamara Pinterich et al.**

We thank the reviewers for their constructive comments. Please find below detailed responses to each comment or question, including notations of improvements to the manuscript. Reviewer comments are *italic*. Changes to the text are in blue font and underlined.

**Anonymous Referee #1 Received and published: 28 June 2017**

This work reports on a new way to quantify aerosol hygroscopic growth factors. The authors introduce the Humidity-controlled water-based Fast Integrated Mobility Spectrometer (HFIMS). The topic of this paper certainly fits in this journal and the paper is very well-written and to the point. The new instrument is a good addition and highly relevant for ambient applications that require fast time resolution. The instrument provides some advantages over other recent instrument designs that conduct growth factor measurements. Characterization tests were performed that show that the instrument can accurately quantify growth factors and deliquescence/efflorescence RH values for a known inorganic salt. The authors can strengthen their manuscript by demonstrating that the instrument works well on a mobile platform. If this is not possible for some reason, the authors should mention why this is not possible; from what i can tell, the field data shown were from a stationary set-up.

Specific Comments:

1. Figure 2: "Sheat flow" spelled wrong (Sheath)

We replaced "Sheat" with "Sheath" in Figure 2 and thank the reviewer for pointing out the misspelling.

2. The authors should provide a clear statement about the range of dry diameters the instrument can handle for the growth factor measurement. is it 35-165 nm as Lines 6-7 suggest on page 6? state it clearly and compare it to other available instruments.

We thank the reviewer for this suggestion. Under the operation conditions in this study, the HFIMS dry particle size range is 15 to 205 nm. Available systems based on optical or aerodynamic sizing (i.e., DASH-SP or H-DMA-APS) are better suited for measuring the hygroscopicity of larger particles. We note that the size range for DMA or WFIMS depends on sheath flow rate and classifying voltage. For example, the upper particle size range could be increased by operating DMA or WFIMS at a lower sheath flow rate. On the other hand, a lower sheath flow rate leads to either a reduced sampling rate (when the ratio of sheath to aerosol flow rate is maintained) or a reduced size resolution (when the aerosol flow rate is maintained). In this study, the WFIMS was operated to cover the smaller size range, in particular, the 5 standard sizes ranging from 35 to 165 nm suggested by the EUSAAR project. Following the

reviewer's suggestion, we now clearly stated the HFIMS' dry particle diameter range, and included the size range of other available instruments for comparison. Following paragraph and table were added to the manuscript (p.5, lines 21 - 26):

"In this study, WFIMS separating voltages ranged from 70 to 4500 V, allowing hygroscopicity measurement for particles with dry diameters ranging from 15 to 205 nm. This includes the standard sizes from 35 to 165 nm suggested by the EUSAAR project (Duplissy et al. 2009). Dry particle size ranges of the HFIMS and other representative instruments are shown in Table 1. Compared to HFIMS, systems based on optical or aerodynamic sizing have a larger upper size limit. On the other hand, HFIMS is capable of rapid measurement of particles with dry diameters below 150 nm, which are difficult to detect using an OPC or APS."

Table 1 Dry particle diameter range  $(D_{p,0}^{\min} - D_{p,0}^{\max})$  of four particle hygroscopicity instruments including HFIMS. Diameter ranges of HFIMS, BMI HTDMA and DASH-SP were estimated based on growth factors ranging from 0.8 to 2.2. DASH-SP dry size range was estimated for 5 l/min DMA sheath flow rate, 281 V minimum classifying voltage and 135 nm cut size of the OPC as specified in Sorooshian et al. (2008). The particle dry diameter range of BMI HTDMA was calculated for a typical fieldwork configuration (i. e.  $Q_{a}$ =0.6 l/min,  $Q_{sh}$ =6 l/min, Lopez-Yglesias et al., 2014) and classifying/scanning voltages ranging from 20 V to 6000 V. The diameter range of H-DMA-APS is listed as specified in Leinert & Wiedensohler (2008).

| Instrument                              | Dminp,0 [nm] | Dmax[nm] |
|-----------------------------------------|-------------------------------------------|----------------------------|
| HFIMS (This study)                      | 15                                 | 205                 |
| BMI HTDMA (Lopez-Yglesias et al. 2014)  | 33                                 | 425                 |
| DASH-SP (Sorooshian et al. 2008)        | 170                                | 455                 |
| H-DMA-APS (Leinert & Wiedensohler 2008) | 800                                | 1600                |

We also highlighted the advantage of DMA-OPC and DMA-APS systems for measurements of larger particles (p. 2 lines 23 – 25):

"These systems, based on optical sizing or aerodynamic sizing, are capable of measuring the hygroscopicity of large accumulation mode particles, which are important to evaluating the optical properties and direct radiative effects of ambient aerosols."

3. It is nice that it can detect smaller sizes where there may be higher concentrations, but it may be useful to point out that larger sizes may be more relevant for light-scattering and radiative forcing? For what purposes are smaller sizes most relevant (e.g., health effects)?

We thank the reviewer for the suggestion and comment. The manuscript was revised accordingly:

P. 2, lines 23 – 25:

"These systems, based on optical sizing or aerodynamic sizing, are capable of measuring the hygroscopicity of large accumulation mode particles, which are important to evaluating the optical properties and direct radiative effects of ambient aerosols."

P. 2, lines 32-34 and p. 3, lines 1 – 2:

Compared to systems based on optical or aerodynamic sizing, HFIMS extends fast measurements to particles with diameters below 150 nm. Particles smaller than 150 nm often represent a large fraction of cloud condensation nuclei population, and may have a strong impact on human health (Chen et al. 2016)."

4. A few times it is stated that "...size distribution spanning a factor of ~3 in particle diameter". Rather than using the confusing "factor of ~3" everywhere, can the authors simply just state the range of diameters?

The size range of humidified particles that can be measured simultaneously by WFIMS depends on the separating voltage. At a constant separating voltage WFIMS can measure size distributions spanning a factor of about 3 in particle diameter (a factor of 10 in electrical mobility) simultaneously. The actual diameter range of the WFIMS is therefore varied according to the diameter of dry particles classified by the DMA.

For example, at a particle dry diameter of 35 nm WFIMS separating voltage is set such that humidified particle with diameters ranging from 28 nm (GF = 0.8) to 95 nm (GF = 2.7) can be measured simultaneously. HFIMS growth factor ranges for all particle dry sizes investigated in this study are given in SI section S4.

5. What is the residence time of aerosol in different parts of the instrument?

Long DMA ( $Q_{sh} = 3 \text{ I/min}, Q_a = 0.3 \text{ I/min}$ )  $\rightarrow 7.4 \text{ s}$ Nano DMA ( $Q_{sh} \rightarrow = 3 \text{ I/min}, Q_a = 0.3 \text{ I/min}$ )  $\rightarrow 0.8 \text{ s}$ RH control unit and tubing between DMA outlet and WFIMS inlet ( $Q_a = 0.3 \text{ I/min}$ )  $\rightarrow 6.5 \text{ s}$ WFIMS  $\rightarrow 1.5 \text{ s} - 2.6 \text{ s}$  depending on particle trajectory

The particle residence time inside the WFIMS depends on particle trajectory. The residence time for each measured particle can be calculated from its detected position and explicitly accounted for (Olfert et al., 2008). We also note that in this initial study, the residence times inside the DMA and between DMA outlet and WFIMS inlet were not optimized. For example, the residence time inside the long DMA could be reduced by operating the DMA at sheath and aerosol flow rates of 10 l/min and 1 l/min, respectively, without negative impact on dry particle size range or sampling rate.

6. How long does it take to scan through a range of different relative humidities? Provide numbers and compare to other instruments.

Hygroscopicity of ambient particles is typically measured by HTDMA at a single RH for a range of particle sizes, similar to results shown in Figure 10. We agree that measurement at multiple RH provides a more complete characterization of particle hygroscopicity. The current RH control unit was not optimized for stepping the RH of HFIMS, such that it takes several minutes for the RH to stabilize at the new setpoints. Optimization of the RH control unit, possibly using a similar design as reported in Lopez – Yglesias et al. (2014) will be included in future improvements.

**7. What is the range of relative humidities the instrument can handle?**

Currently stable RH conditions can be maintained from 20% - 90% as shown in Figure 5.

8. Section 2.1: Can the authors comment on hysteresis effects that are common with nafion based systems?

The RH of aerosol sample exiting the Nafion exchanger is measured by an RH sensor immediately down-stream of the exchanger, and the RH measurement is input to a PID module, which controls the aerosol sample RH (Fig. 1). Therefore, we expect the potential hysteresis effects of the Nafion exchanger has negligible impact on aerosol sample RH.

9. Has the instrument been deployed on a mobile platform? The paper states that the instrument is ideal for mobile platform work but it is unfortunate that data does not appear to be shown from its use in the field to give readers a better sense of how robust it is when the conditions aren't 'easy' such as in a lab or at a stationary field site.

The instrument has not (yet) been deployed on a mobile platform. We removed the part '... or measurements onboard mobile platforms.' from the last sentence of the conclusions section (p. 9, lines 24 - 25).

10. Figure S2: x-axis has "distance" misspelled.

We replaced "Distance" with "Distance" in Figures S2 (a) and (b) and thank the reviewer for pointing out the misspelling.

**Anonymous Referee #2 Received and published: 28 August 2017**

A new instrument for fast measurement of particle hygroscopicity, HFIMS, is reported in this paper. Different from the existing fast hygroscopicity measurement instruments, HFIMS measures the electrical mobility diameters of the grown particles so that particle density and refractive index are not needed anymore. Compared with the HTDMA, HFIMS avoids the diameter scan which is time consuming therefore has a much higher time resolution and may work on mobile platforms. The topic fits well in AMT and the manuscript is well written. I therefore recommend the final publication of this paper on AMT.

**Specific comments:**

1. What is the maximum RH the system can reach? How about its stability at different RH levels? Since the RH is only measured before the flow entering WFIMS (not like HTDMA in which RH is also measured at the sheath exit), the real RH the particles undergo may be slightly different from the value given by RH sensors (e.g. in case of high particle concentration and very slow growth, or temperature difference between parts in the system). Does the author have any comment on this?

With the current RH control unit the maximum RH that the system can hold stable is 90%. As pointed out by the reviewer it might be possible, that the RH during mobility classification slightly differs from the measured sheath/aerosol RH. However, data presented in Fig. 5 suggest that these differences are minor. Thermal insulation of the RH control section and WFIMS separator could further reduce temperature differences between relevant parts, and will be part of future improvements.

To answer the reviewer's comment on the effect of water vapor depletion, we estimated the change in RH due to water uptake at standard temperature and pressure for a growth factor of 2 and dry particle sizes up to 200 nm. For DMA classified ambient particles with concentrations up to  $10^4$  cm-3 the change in RH (initially 85%) due to water vapor uptake inside the WFIMS separator should be less than 0.1‰.

2. What is the time needed to washout all particles in the system at a typical setting of flow rates? The washout time may strongly influence the time resolution of the measurement.

Figure R 1 Time evolution of particle number concentration  $N_{FIMS}$  measured by WFIMS (black line) showing a steep concentration drop after setting nano DMA voltage to 0 V (t ~ 21.875 h). Data around the concentration drop were fitted using Eq. (R1) to determine the half-life  $\tau_{1/2}$  for concentration decay (Eq. R2).

$$N_{FIMS}(t) = N_0 \cdot e^{-\lambda \cdot t} \tag{R0}$$

$$\tau_{1/2} = \frac{\ln(2)}{\lambda} \tag{R0}$$

In this study the washout time was determined from the response of FIMS after setting the nano DMA voltage to 0 V, which creates a nearly stepwise change of classified particle concentration down to 0 cm-3. Particle concentrations measured by WFIMS, i.e. NFIMS (black line in Fig. R1), were fitted (red line in Fig. R1) around the steep drop (t ~ 21.875 h) using an exponential decay model (Eq. R1). The half-life  $\tau_{1/2}$  (Eq. R2) of particles within the system was found to be 1.1 s. In other words it takes the HFIMS about 4.4 seconds (4·  $\tau_{1/2}$ ) to washout 94% of all particles.

3. The finite-width transfer function of WFIMS may cause a smooth effect in the measured distribution of GF. The discussion about it (P7L10) is a bit too brief. Can the author give a more detail discussion on the uncertain of GF distribution due to this effect? Is it possible and necessary to correct this smoothing effect?

As the reviewer pointed out correctly, the measurement signal of the HFIMS is a smoothed integral transform of the particle's actual growth factor probability distribution function (GF-PDF) together with the transfer functions of the DMA and WFIMS. Hence, a data inversion algorithm should be applied to the response signal of the HFIMS measurements to retrieve the GF-PDF. Because DMA transfer function, WFIMS transfer function, and the response of the HFIMS are known (Stolzenburg & McMurry, 2008; Kulkarni and Wang, 2006), the actual GF-PDF could be solved by the data inversion using an algorithm similar to TDMAfit (Stolzenburg & McMurry, 1988, 2008) and TDMAinv (Gysel et al., 2009). The development and demonstration of this algorithm are part of an ongoing study.

- 15 with the WFIMS, the HFIMS greatly increases the speed of particle growth factor measurement. The performance of the HFIMS was evaluated using NaCl particles with well-known hygroscopic growth behavior, and further through measurements of ambient aerosols. Results show that HFIMS can reproduce, within 2 % the literature values for hygroscopic growth of NaCl particles. NaCl deliquescence was observed between 76 % and 77 % RH in agreement with the theoretical value of 76.5 % (Ming and Russell, 2001), and efflorescence relative humidity (43 %) was found to lie within
- 20 the RH range of 41 % to 56 % reported in the literature. Ambient data indicate that HFIMS can measure the hygroscopic growth of five standard dry particle sizes ranging from 35 to 165 nm within less than three minutes, which makes it about an order of magnitude faster than traditional HTDMA systems.

**1** Introduction**

25

5

The hygroscopicity of atmospheric aerosols is a key parameter in determining their impact on global climate. The uptake of water by individual particles increases the light scattering, enhances heterogeneous chemical transformations important to secondary aerosol formation (e.g., Surratt et al., 2010), and is important in the formation of cloud droplets. The abundance of hygroscopic particles that act as cloud condensation nuclei affects cloud formation and cloud droplet number concentrations, which in turn influences cloud albedo, coverage and lifetime (Twomey 1977; Albrecht 1989). These "indirect effects" of atmospheric aerosols on the Earth's radiation balance remain one of the largest uncertainties in understanding climate change (IPCC 2013). Hygroscopicity is among the key determinants of the ability of aerosol particles to form cloud droplets and therefore the aerosol indirect effects (e.g. Mei et al., 2013; Liu and Wang, 2010).

Most commonly particle hygroscopic growth is measured using hygroscopicity tandem differential mobility analyzer (HTDMA) systems, which consist of two differential mobility analyzers (DMAs) in series, separated by a means to control

- 5 the sample flow relative humidity (RH). HTDMA systems first select a single particle size using the first DMA, change its relative humidity environment, then scan the classifying voltage of the second DMA to measure the distribution of particle sizes resulted from the change in RH. The HTDMA method is accurate, but slow. Typically the time required to complete a measurement cycle of determining the growth factor at a single relative humidity (such as 90 %) for 5 different particle sizes is about 30 min (e. g. Cerully et al., 2011). Measurement periods are especially long for large particles which are low in 10 concentration, and small particles that have a low charging efficiency.
- Several investigators have worked to increase the speed of HTDMA measurements by replacing the second DMA with an instrument that is capable of fast size distribution measurements. Sorooshian et al. (2008) developed a Differential Aerosol Sizing and Hygroscopicity Spectrometer Probe (DASH-SP), in which wet particle size is measured by an optical particle counter (OPC). By replacing the 2nd DMA with an optical counter, DASH-SP accelerates the measurement significantly.
- 15 However, the optical counting limits DASH-SP measurements to particles larger than ~150 nm in diameter, and a sophisticated algorithm is required to account for the variation of particle refractive index due to water uptake and its impact on optical sizing. Stolzenburg et. al (1998) developed a high-flow mobility analyzer which they coupled to an optical particle counter, with humidity control upstream of the mobility size separation, and an aerosol dryer downstream to measure particle size change upon dehumidification. Leinert and Wiedensohler (2008) developed a DMA-aerodynamic particle sizing (APS)
- 20 system to examine growth factors, but their measurements were complicated by the change in particle density, which affects the aerodynamic measurement. While faster, the limitation of these coupled DMA-optical counter or DMA-aerodynamic sizing techniques is the limitation on the measurement size range and the additional complexity or uncertainty in mapping the optical or aerodynamic size onto the physical size of the particle that is important to assessing water uptake. These systems, based on optical sizing or aerodynamic sizing, are capable of measuring the hygroscopicity of large accumulation
- 25 mode particles, which are important to evaluating the optical properties and direct radiative effects of ambient aerosols. To address the need for fast and precise measurements of particle hygroscopic growth, we have developed a Humiditycontrolled water-based Fast Integrated Mobility Spectrometer (HFIMS), which replaces the second DMA of the HTDMA systems with a water-based FIMS (WFIMS; Pinterich et al., 2017). By detecting particles of different sizes simultaneously, WFIMS provides rapid measurements of the size distribution of humidified particles. Unlike the final optical sizing of
- 30 Sorooshian et al. (2008), or final aerodynamic sizing of Leinert and Wiedensohler (2008), WFIMS measures particle sizes based on electrical mobility. This removes the uncertainty introduced by particle refractive index or density, and provides the same, precise growth factor measurements of the HTDMA systems, but with a much faster measurement speed. Importantly, the HFIMS allows hygroscopicity measurements in the critical size range below 100 nm, which generally dominates atmospheric particle number concentrations. 
[revised manuscript text omitted]

---

## Author Comment (AC2) · 16 Oct 2017

We thank the reviewer for constructive comments. Detailed responses to each comment or question, including notations of improvements to the manuscript were uploaded in the form of a supplement.

Please also note the supplement to this comment:
https://www.atmos-meas-tech-discuss.net/amt-2017-180/amt-2017-180-AC2-supplement.pdf